# My House, My Rules: Learning Tidying Preferences with Graph Neural Networks

**Ivan Kapelyukh**
The Robot Learning Lab
Imperial College London
ik517@imperial.ac.uk

**Edward Johns**
The Robot Learning Lab
Imperial College London
e.johns@imperial.ac.uk

**Abstract:** Robots that arrange household objects should do so according to the user's preferences, which are inherently subjective and difficult to model. We present NeatNet: a novel Variational Autoencoder architecture using Graph Neural Network layers, which can extract a low-dimensional latent preference vector from a user by observing how they arrange scenes. Given any set of objects, this vector can then be used to generate an arrangement which is tailored to that user's spatial preferences, with word embeddings used for generalisation to new objects. We develop a tidying simulator to gather rearrangement examples from 75 users, and demonstrate empirically that our method consistently produces neat and personalised arrangements across a variety of rearrangement scenarios.

**Keywords:** graph neural networks, preference learning, rearrangement tasks

## 1 Introduction

Rearranging objects in unstructured environments is a fundamental task in robotics. For example, a domestic robot could be required to set a dinner table, tidy a messy desk, and find a home for a newly-bought object. Given goal states stipulating where each object should go, many approaches exist for planning a sequence of actions to manipulate objects into the desired arrangement [1, 2, 3].

But how does a robot know what these goal states should be, for a rearrangement to be considered successful, tidy, and aesthetically pleasing? The ability of a robot to understand human preferences, especially regarding the handling of personal objects, is a key priority for users [4]. Several factors involved in spatial preferences are shared across many users: symmetry and usefulness are preferable to chaos. Stacking plates neatly inside a cupboard rather than scattering them randomly across a sofa is common sense. However, many of the factors involved are inherently personal, as is their relative prioritisation. For example, is the person left or right-handed? Do they want their favourite book tidied away neatly on a shelf, or placed on the table nearby for convenience? How high is their risk tolerance — do they deliberately keep fragile glasses further from the edge of the shelf, even if it makes them harder to reach? Do they order food in the fridge with the tallest objects at the back and the shortest at the front? Or do they keep the most frequently used items at the front and easily reachable? In this case, how does the robot know which items are most frequently used and ought to be placed near the front? This in itself depends on personal preference.

It is clear that spatial preferences are a complex, multi-faceted, and subjective concept. In this paper, we propose an object rearrangement strategy which models these preferences with a latent user vector representation, inferred by directly observing how a user organises their home environment. The network is trained on arrangements made by a set of training users. For a test user, the robot will observe how they have arranged their home. This can be achieved by a robot using existing techniques for localisation, object identification and pose estimation [5, 6, 7]. By passing these example scenes through the Variational Autoencoder, the network will extract this test user's preference vector. This can then be used by a robot to tidy up these scenes, and also predict personalised arrangements for new objects and new scenes based on how training users with "similar" preferences to this test user arranged them. These predicted arrangements can then be used as target states for planning and manipulation algorithms [1].

5th Conference on Robot Learning (CoRL 2021), London, UK.

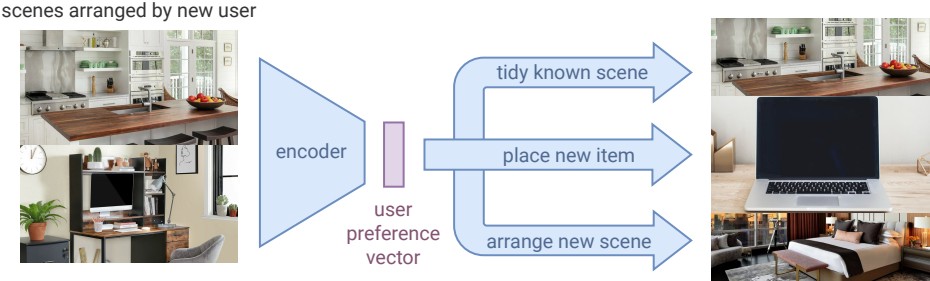

scenes arranged by new user

encoder

user preference vector

tidy known scene

place new item

arrange new scene

Figure 1: NeatNet infers a user preference vector, and can then predict personalised arrangements.

We make the following contributions, adding a layer of personalisation to robotic rearrangement:

**Novel Graph VAE Architecture**. We present NeatNet: an architecture for learning spatial preferences. Users are encoded through representation learning with Variational Autoencoders. Graph Neural Network (GNN) layers are used to learn from scenes arranged by users.

**Word Embeddings for Object Generalisation**. Our network extracts semantic features for an object from the word embedding of its name, enabling generalisation to objects unseen during training.

**User-Driven Evaluation**. We develop TidySim: a 3D simulator web app for users to arrange scenes. With rearrangement data from 75 users, we demonstrate empirically that our method can learn spatial preferences, and predict personalised arrangements which users rate highly. Visualisations of predicted arrangements are available in the supplementary material and on our project page, together with our code and a summary video: www.robot-learning.uk/my-house-my-rules.

## 2    Related Work

A crucial challenge in the field of robotic rearrangement is predicting a position for each object in order to form a tidy arrangement. Prior approaches discretise the problem by semantically grouping objects into containers [8, 9], learn Dirichlet Process models of common object locations [10, 11], or optimise a cost function based on one user's examples [12]. We now discuss these three approaches, and then briefly reference existing neural methods for learning non-spatial user preferences.

**Grouping semantically related objects.** Collaborative filtering [8] can predict how a user would categorise an object based on how similar users categorised it. Data is mined from object taxonomies such as shopping websites to generalise to new objects: preferences about known objects nearby in the hierarchy are used to select a shelf for the new object. Object taxonomies can also be used to place an object into the correct cupboard, viewed as a classification task [9]. While these approaches are effective in grouping objects into boxes or cupboards, our method addresses the general case of learning continuous spatial relations, like how to set a dining table or tidy an office desk.

**Dirichlet Process models**. Recent work [10] learns *spatial concepts*: i.e. the relationship between an object, its tidied location, and a name to describe that location (useful for voice commands). Model parameters are learned with Gibbs Sampling. Another approach with Dirichlet Process modelling uses context from sampled human poses [11] to position objects conveniently. Density function parameters are shared across objects of the same type. This approach is evaluated on a training dataset of arrangements made by 3-5 users, with a further 2 users scoring arrangements. While these methods predict generally tidy arrangements, it is not shown whether they can tailor them to individual preferences. We develop a tidying simulator to gather data on a larger scale, demonstrating that our neural encoder makes personalised predictions on a dataset of 75 users. Furthermore, our method uses word embeddings to predict positions for objects unseen during training, and learns transferable representations of tidying preferences which can be applied to arrange new scenes.

**Example-driven tidiness loss function.** Another approach is to ask each user to provide several tidy arrangements for a scene, to serve as positive examples [12]. To tidy a scene, the robot computes a target tidy layout using a cost function which encourages altering the object-object distances in the untidy scene to match those in the "closest" tidy positive example. Simulated re-annealing is used to optimise this cost function efficiently. We add value firstly because our method generalises to unseen objects without requiring users to arrange them manually, and secondly because our method

combines knowledge about the current user's preferences with prior knowledge about "similar" training users, correcting errors in individual examples and allowing for stronger generalisation.

**Neural recommender systems.** Neural networks have been used for making recommendations based on user preferences [13]. The YouTube neural recommender system [14] predicts a user's expected watch time for a future video based on past videos watched and searches made. However, these methods do not address the challenges of learning spatial relations by observing scenes and predicting tidy arrangements for any set of objects. We apply GNN components to solve this.

# 3 Method

We introduce NeatNet: a novel architecture for learning spatial preferences. First, we describe a Variational Autoencoder for learning latent representations of a user's preferences (Section 3.1). Then we introduce the GNN layers which allow NeatNet to learn from scenes arranged by users, modelled as graphs of objects (Section 3.2). Next, we adapt the architecture to learn from multiple example scenes per user (Section 3.3). Finally, we illustrate how object semantic embeddings are inferred from word embeddings, for generalisation to unseen objects (Section 3.4).

## 3.1 Encoding User Preferences with VAEs

Our objective is to learn a user encoder function $\mathcal{E}_\phi$, represented by a neural network with learned parameters $\phi$. Its output is a user's preference vector $u$. Its input is a representation of a scene arrangement made by that user. Each scene is defined by the terms $s$ (the semantic identities of the objects in the scene, further explained in Section 3.4), and $p$ (the position encodings for those objects, e.g. coordinates). These are not passed into the VAE directly: scenes are first encoded using GNN layers (Section 3.2), since scenes can have a variable number of objects. We also train a position predictor function $\mathcal{D}_\theta$. Given a set of objects identified by $s$, and a user preference vector $u$, this predicts a position for each object which reflects those spatial preferences.

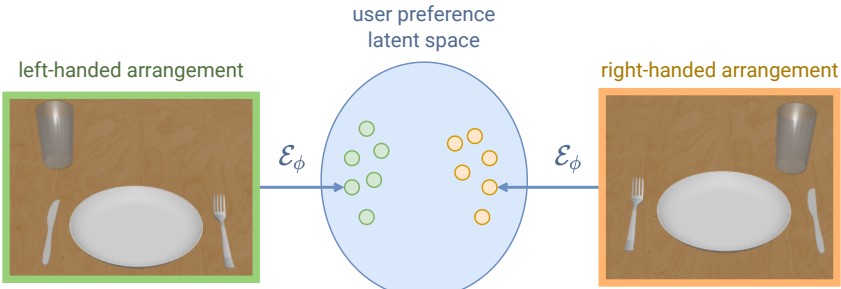

Figure 2: Two users being mapped to points in the user latent space based on how they arrange a dining scene. Users close together in this latent vector space share similar spatial preferences.

These networks are trained end-to-end as a Variational Autoencoder [15]. A scene represented by $s$ and $p$ is passed into the encoder $\mathcal{E}_\phi$. The output user preference vector $u$, along with the semantic embeddings of the objects $s$, is passed into the position predictor $\mathcal{D}_\theta$, which acts as a decoder by reconstructing the original scene. Since the user preference vector is a low-dimensional bottleneck, this encourages the network to learn important spatial preferences like left/right-handedness which are useful for predicting the positions of many objects at once. For variational inference, the encoder $\mathcal{E}_\phi$ predicts a Gaussian distribution over the possible user vectors $u$ which could explain the input scene. At training time, we sample from this distribution to obtain $u$, and we take the mean at inference time. The VAE objective is thus $\phi^*, \theta^* \triangleq \operatorname{argmin}_{\phi,\theta} L(\phi, \theta)$.

$$L(\phi, \theta) \triangleq \mathbb{E}_{p_{data}(s,p)} \left[ \|\mathcal{D}_\theta(\underbrace{\mathcal{E}_\phi(s, p)}_{=u}, s) - p\|_2^2 + \beta KL\left[q_\phi(u|s,p)\|p(u)\right] \right] \tag{1}$$

Here, $p_{data}$ represents the training data distribution. $p(u)$ is a prior over the user preference vectors: a standard Gaussian, with zero mean and unit variance. KL represents the Kullback–Leibler divergence (standard in VAEs [15]) which encourages the estimated $u$ distributions to stay close to

the $p(u)$ Gaussian prior over the user space, and the hyperparameter $\beta$ scales this prior term to make the latent space less sparse and encourage disentanglement [16].

## 3.2 Encoding Scenes with GNN Layers

We will now encode scenes as vectors so that they can be passed through the VAE. To do this, we use Graph Neural Network (GNN) layers. GNNs can operate on scenes with a variable number of arbitrary nodes, which is critical since the robot cannot foresee all the objects it will ever encounter. Each node represents an object in the scene. Node $i$ has feature vector $x_i$ formed by concatenating the semantic embedding for that object $s_i$ with its position encoding $p_i$. The scene graph is fully connected, to avoid making assumptions about which object-object relations are relevant or not.

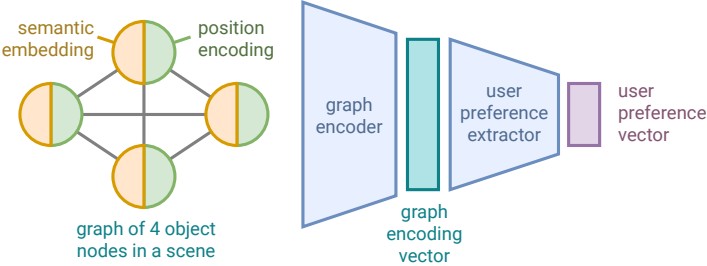

Figure 3: The encoder network, all together representing the function $\mathcal{E}_\phi$ defined in Section 3.1. The input is a node feature matrix with a row for each node's feature vector $x_i \triangleq s_i \| p_i$. The graph encoder consists of GAT layers (described later) separated by ELU non-linearities [17]. Then we pass the final feature vector of each node through a network of fully-connected layers, which has the same output size as the graph encoding vector. To aggregate the output node vectors into a single graph encoding vector, we apply a global add-pooling layer. This graph encoding vector is passed through a user preference extractor: a series of fully-connected layers separated by LeakyReLU activations [18]. The output is a mean and variance for the user preference vector distribution, from which we sample to obtain $u$. This is a latent vector which may not always be interpretable, but it might capture features such as whether the user prefers convenience over aesthetics.

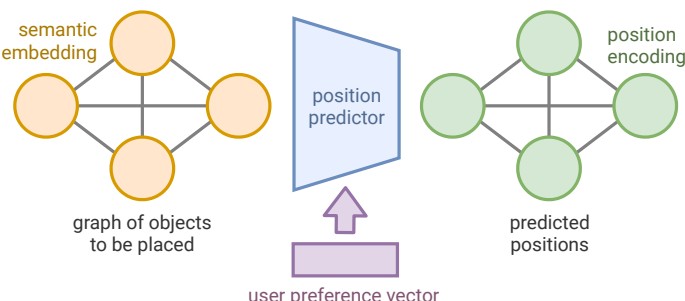

Figure 4: The position predictor, which acts as the decoder $\mathcal{D}_\theta$. The input now consists of a set of objects for which we wish to predict positions, according to a user preference vector $u$. We set the node feature vectors to be $x_i \triangleq p_i \| u$. This time, the output is a position vector for each node. Our experiments focus on positions, but the position vector can be extended to include orientation information. Similarly to the graph encoder, the position predictor consists of a sequence of GAT layers, after which each node is passed through a series of fully-connected layers, the final output of which is the predicted position vector for that node.

Both the graph encoder (Figure 3) and position predictor (Figure 4) use GNN layers. As is standard for GNN layers, we need to compute the hidden feature vector $h_i$ for each node $i$, based on its current feature vector $x_i$ as well as the node feature vectors $x_j$ of its neighbours. To compute $h_i$, we choose the Graph Attention (GAT) layer [19]. $h_i$ is computed as a weighted sum of the node feature vectors $x_j$ in the current node's neighbourhood, where the weights $\alpha_{ij}$ are generated by the soft

attention mechanism. Since our graph is fully connected, we calculate attention weights for all the other objects in the scene: therefore the attention mechanism allows the network to autonomously decide which object-to-object relations are the most important for predicting correct arrangements. In order to further increase the power of the graph network, we can add a second GAT layer which takes as input the $h_i$ features produced by the first layer. A GAT layer computes $h_i$ as follows, where $\|$ represents concatenation, and the learned parameters are the weights $\mathbf{W}$ and $a$:

$$h_i \triangleq \sum_{j \in \mathcal{N}_i} \alpha_{ij} \mathbf{W} x_j \qquad \alpha_{ij} \triangleq \frac{\exp\left(\text{LeakyReLU}\left(a^T \left[\mathbf{W} x_i \| \mathbf{W} x_j\right]\right)\right)}{\sum_{k \in \mathcal{N}_i} \exp\left(\text{LeakyReLU}\left(a^T \left[\mathbf{W} x_i \| \mathbf{W} x_k\right]\right)\right)} \qquad (2)$$

### 3.3 Generalising Across Scenes

Many spatial preferences, such as left/right-handedness, will be exhibited by the user consistently across scenes. We now extend NeatNet to learn from multiple scenes arranged by the same user, as shown in Figure 5. Every example scene arranged by the current user is passed through the single-scene encoder network $\mathcal{E}_\phi$ in Figure 3. Each scene may produce a subtly different estimate of the user's preferences, so they are aggregated through a global pooling layer to produce an average user preference vector $u$. On the decoding side, we use the same $u$ vector as the input to the position predictor to reconstruct *all* the scenes created by this user. Since $u$ is the architecture's bottleneck, this encourages the network to learn important preference features which generalise across scenes, such as left/right handedness or a preference for arranging objects compactly rather than spaciously.

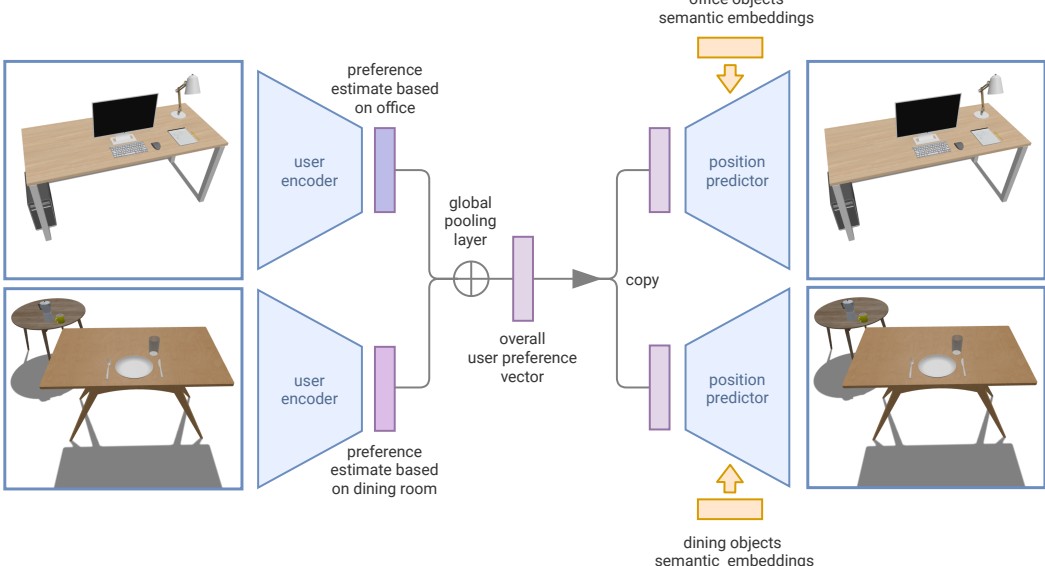

Figure 5: Learning one generalised user preference vector across multiple example scenes per user.

### 3.4 Semantic Embeddings

Each object's identity is represented by its semantic embedding vector. One-hot encoding is a straightforward approach. A limitation of this method is that it does not generalise readily to new objects. An alternative is feature vectors: example entries include the height of an object, or RGB values describing its colour palette. However, this may not capture all features necessary for tidying. We also develop a method using word embeddings. Objects which appear in similar linguistic contexts are used for similar activities, and so are often arranged in a related way: for example "pen" and "pencil", or "salt" and "pepper". We load word embeddings from a FastText model [20], pre-trained on the Common Crawl dataset of over 600 billion tokens. Names of objects are provided to NeatNet either by the simulator or by an object detection system if deployed physically. These names are converted to word embeddings and passed through the semantic embedding extractor (Figure 6), yielding the semantic embeddings referred to in Figure 3. This allows the robot to predict positions for new objects never seen before in any tidy arrangement during training.

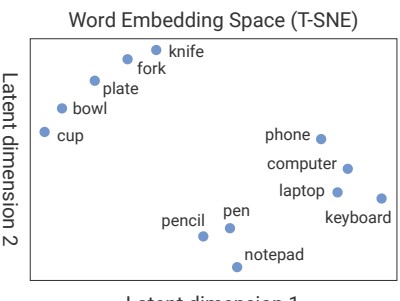
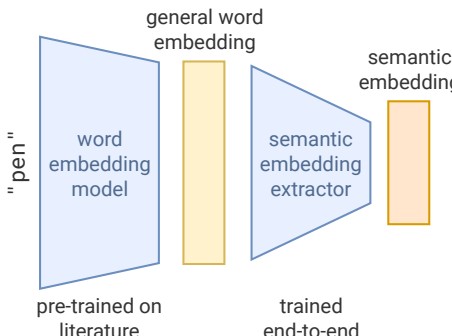

Figure 6: **Left:** Word embedding space. Object names which appear in similar linguistic contexts will be close together in this latent space. **Right:** the semantic embedding extractor outputs an object's semantic embedding vector, using fully-connected layers. This is trained end-to-end with NeatNet, to extract the object features which are most relevant for our spatial arrangement task.

# 4 Experiments

## 4.1 Collecting Rearrangement Data from Users

Gathering data from human users at scale is a significant challenge for robotics experiments. We developed TidySim: a 3D object rearrangement simulator deployed as a web app, allowing users to arrange a series of scenes through an intuitive interface. Features include: moving the camera, clicking and dragging objects to move or rotate them, and an inventory for unplaced objects.

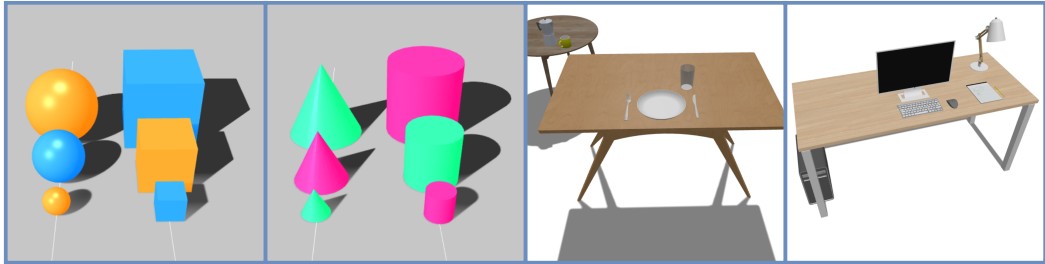

Figure 7: Scenes arranged by users. From left: (1-2) are abstract scenes, where users must split the objects into two lines, either by shape or by colour, (3) is a dining room, and (4) is an office desk.

## 4.2 Experimental Procedure

We designed the scenes in Figure 7 to maximise experimental value while keeping the user workload low ($\sim$ 5 minutes total per user). Rearrangement data from 75 users was gathered by distributing the experiment link on social media. Each user submits a tidy arrangement for a number of scenes. We set aside a group of 8 test users, male and female, and majority non-roboticists. The training dataset was formed from the 67 remaining users. NeatNet was trained by passing training users through the Graph VAE and updating parameters with a neural network optimiser according to the loss function in Equation 3.1. For each test user, we passed their example scenes through NeatNet to extract their spatial preferences, and predict a personalised arrangement for the test scene (varies by experiment). Arrangements were also produced by the baseline methods in Section 4.3. The test user was then asked to score the arrangements produced by each method based on how tidy they were according to their preferences, on a scale from 0 (completely untidy) to 10 (perfectly tidy).

## 4.3 Baselines

In Section 2, we qualitatively compare with existing methods to show where we add new capabilities. For a fair quantitative evaluation of our method on this dataset, we test against a variety of baseline methods depending on the specific rearrangement task. `Positive-Example`: to tidy a known

scene, this baseline copies the example arrangement which was supplied by the test user (which the `NeatNet` method uses to extract this user's tidying preferences). This baseline ignores priors from other users and is unavailable when generalising to new objects or scenes. `Random-Position` sets the missing object's predicted position randomly, while `Mean-Position` uses the object's mean position across all the example arrangements from training users. `NeatNet-No-Prefs` is an ablation baseline: the test user's preference vector is replaced with the zero vector (representing a neutral preference, which test users might not find tidy). The `Pose-Graph` method produces an arrangement which some users will find tidy, but is still not personalised to the current user's preferences. It learns a probabilistic pose graph of relative object positions from the training example arrangements, and solves it using Monte Carlo sampling and scoring to find the most likely arrangement. When placing an unseen object, `Mean-With-Offset` uses the mean position of the other objects in the scene, and adds a small random offset to minimise collisions with other objects. `Nearest-Neighbour` finds the object already placed by the user which has the closest word embedding vector to the unseen object, and places the new object next to it, i.e. it would place the salt next to the pepper. `Weighted-kNN-Regression` uses the $k = 3$ nearest neighbours, weighted by word distance, to compute an average position. For the task of arranging a new scene, the `Random-User` baseline copies how a random training user arranged that scene, without personalisation. `kNN-Scene-Projection` projects the user's preferences from the example scene onto the new scene, by placing each object in the new scene based on how the user placed the $k = 3$ most similar objects in the example scene, similar to the `Weighted-kNN-Regression` algorithm.

## 4.4 Experiment Results

We designed a series of experiments to test our method's capabilities. Visualisations of predicted arrangements are available in the supplementary material and on our project page: www.robot-learning.uk/my-house-my-rules.

**Can NeatNet Tidy a Known Scene?** (Table 1) Suppose that the robot has seen the user tidy a room. After the room becomes untidy, the robot must tidy up each object. This tests how well the network can reconstruct the scene based on the user preference vector and its prior knowledge. For each scene separately, we trained NeatNet as described in Section 4.2. Then for each test user we pass their scene through the VAE and ask them to rate the reconstructed scene.

| Tidying Method | Abstract 1 | Abstract 2 | Dining Table | Office Desk | Average |
|:---:|:---:|:---:|:---:|:---:|:---:|
| `Mean-Position` | 3.25 ±2.04 | 3.00 ±1.94 | 2.34 ±1.17 | 2.22 ±1.48 | 2.70 ±1.50 |
| `NeatNet-No-Prefs` | 4.50 ±1.73 | 3.75 ±1.31 | 2.58 ±1.06 | 3.70 ±1.10 | 3.63 ±1.38 |
| `Pose-Graph` | — | — | 6.98 ±1.53 | 7.82 ±0.86 | 7.40 ±1.22 |
| `Positive-Example` | **8.75** ±1.26 | 7.50 ±0.55 | 8.60 ±0.58 | **9.18** ±0.81 | 8.51 ±0.98 |
| `NeatNet` | **8.75** ±1.46 | **8.25** ±1.45 | **9.12** ±0.49 | 8.90 ±0.40 | **8.76** ±1.17 |

Table 1: Tidying a known scene. Rows are tidying methods and columns are scenes. The mean and standard deviation of tidiness scores given by test users are shown, along with a cross-scene average.

`NeatNet` can reconstruct personalised tidy arrangements from a user preference vector, outperforming baselines and approaching the user's handmade `Positive-Example`. In a direct comparison, all test users ranked `NeatNet` over `Pose-Graph`. Surprisingly, users sometimes even prefer the reconstructed tidy scene *over their own example*. This is because in each individual user-created arrangement, there is noise in how they placed each object (realistically this would also include error from localisation/pose estimation). Our method uses prior knowledge from other users to correct for noise in individual examples (analogous to VAE image denoising [21]).

**Can NeatNet Find a Home for a New Object?** (Table 2) If a user brings a new object (e.g. a cup) into the scene, the robot should predict a tidy position for it. NeatNet is trained as before, but with the positions of random nodes masked out from input examples to encourage generalisation. At test time, we mask out the new object's position from the test user's example arrangement, but the position predictor must still place it. This is analogous to the use of generative models for "inpainting" missing or obstructed regions of input images [22]. This experiment is repeated for each object. `NeatNet` outperforms the baselines by successfully combining what it knows about the current user's preferences with prior knowledge of how similar users placed the new object.

| Prediction Method | Cup | Fork | Knife | Plate | Average |
|---|---|---|---|---|---|
| Random-Position | 26.70 ±9.07 | 15.76 ±5.51 | 15.96 ±8.03 | 7.10 ±4.49 | 16.38 ±8.19 |
| NeatNet-No-Prefs | 13.20 ±6.52 | 14.98 ±5.01 | 13.38 ±5.77 | 2.88 ±1.02 | 11.12 ±6.23 |
| Mean-Position | 11.42 ±5.19 | 15.06 ±5.50 | 13.48 ±6.27 | **1.64** ±0.65 | 10.40 ±6.14 |
| NeatNet | **5.72** ±1.46 | **1.18** ±0.76 | **1.44** ±1.15 | 1.96 ±0.67 | **2.58** ±1.34 |

Table 2: Placing a new object into the scene. Figures show the mean and standard deviation of the absolute error between the predicted and true, unmasked positions of each object (in centimetres).

**Can NeatNet Generalise to Objects Unseen During Training?** (Table 3) The network must place a new blue box into an abstract scene, and a laptop into the office scene, though it has never seen a laptop during training. The training process is similar to the previous experiment. NeatNet's placement of the new objects (box and laptop) is satisfactory to users. To place the laptop, NeatNet considers how the user placed several semantically similar objects: the computer, laptop and keyboard. The Nearest-Neighbour baseline places the laptop based solely on how the user placed their desktop computer, thus misplacing it under the desk. Weighted-kNN-Regression performs well, but NeatNet was still ranked higher by every user because it can extract the semantic information from the word vectors which is most useful for tidying, since the network is trained end-to-end. Furthermore, NeatNet is able to learn and extrapolate the order-of-size pattern in the abstract scene.

| Tidying Method | Abstract Scene | Office Desk | Average |
|---|---|---|---|
| Mean-With-Offset | 2.58 ±1.35 | 4.68 ±1.31 | 3.63 |
| Nearest-Neighbour | 5.78 ±1.32 | 1.70 ±0.67 | 3.74 |
| Weighted-kNN-Regression | 4.22 ±0.83 | 7.80 ±1.64 | 6.01 |
| NeatNet | **8.10** ±1.24 | **9.16** ±1.28 | **8.63** |

Table 3: Placing an object unseen during training, with user scores for tidiness shown.

**Can NeatNet Predict a Personalised Arrangement for a New Scene?** (Table 4) At training time, the network is shown how the user has arranged both abstract scenes, with random scene masking applied to encourage generalisation. At test time, the network is shown one abstract scene and asked to predict how the test user would arrange the other. kNN-Scene-Projection produces personalised arrangements, but NeatNet lines up objects more neatly because it combines personalisation with learned prior knowledge about these objects from training users. The Random-User baseline performs moderately well, especially when by luck the random training user which was chosen has similar preferences to the test user. However, when those preferences differ, the neural network method out-performs it. This shows the importance of accounting for preferences, and demonstrates that the network can learn preferences which enable generalisation across scenes.

| Tidying Method | Abstract 1 | Abstract 2 | Average |
|---|---|---|---|
| Mean-Position | 2.04 ±1.05 | 2.30 ±0.97 | 2.17 |
| Random-User | 5.52 ±1.12 | 6.24 ±1.67 | 5.88 |
| kNN-Scene-Projection | 6.56 ±0.96 | 7.32 ±1.48 | 6.94 |
| NeatNet | **9.58** ±0.53 | **9.72** ±0.44 | **9.65** |

Table 4: User scores for predicted arrangements of a new scene.

## 5   Conclusions

**Findings.** Our NeatNet architecture learned latent representations of user spatial preferences by training as a Variational Autoencoder. Graph Neural Network components allowed it to learn directly from user-arranged scenes, represented as graphs, and word embeddings enabled predictions for unseen objects. NeatNet's ability to generalise and make personalised predictions was demonstrated through experiments with real users, adding new capabilities to existing methods.
**Future work.** Rather than using the word embedding of the object's name, convolutional layers can be used to generate a semantic embedding for an object based on its visual appearance (e.g. teacups look similar to mugs and are placed similarly). Furthermore, we can combine our approach with model-based priors from existing methods, e.g. using human pose context [11] as part of our loss function to ensure objects are placed conveniently, while also tailoring to individual preferences.

**Acknowledgments**

This work was supported by the Royal Academy of Engineering under the Research Fellowship scheme.

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
