# OpenReview forum: "My House, My Rules: Learning Tidying Preferences with Graph Neural Networks"
_robot-learning.org/CoRL/2021/Conference — CoRL2021 Poster_

### Official Review · Reviewer_Ftgd · 2021-07-22

**Originality:** Good
**Technical Quality:** Good
**Clarity Of Presentation:** Excellent
**Impact:** 4

**Recommendation:**

Strong Accept: I recommend accepting the paper and will argue for my recommendation even if other reviewers hold a different opinion.

**Summary:**

This work provides a method to enable a robot arrange/tidy objects according to a user’s preferences. NeatNet is a variational autoencoder that learns a representation of the user’s preferences and uses it to position objects in a scene. Input scenes are encoded using graph neural network layers.



**Issues:**

- Addition of stronger baselines (see comments above)
- Providing more examples of both experimental results as well as different user preferences.
- Including standard deviation for all means in all tables.

**Reviewer Expertise:**

Good: General knowledge of the area

**Strengths And Weaknesses:**

The paper is clearly written and organized. The authors propose a novel scheme to learn user preferences of spatial object structures by combining graph neural network layers with a variational autoencoder architecture. The series of questions answered in the evaluation section seem comprehensive.

However, the paper could be strengthened by adding baselines that also model the user’s preferences. Currently, none of the baselines model user preferences which provides for an unfair comparison. For instance, in Tables 1 and 4, it’s not surprising that the rest of the baselines (Mean-Positions, NetNet-No-Prefs, and Pose-Graphs, Mean-Position, Random-User) would do poorly. Adding baselines that take into account user preferences would also shed light on why using GNN layers + VAE the way the paper did is the best way to learn the user’s spatial preferences rather than doing something like inverse reinforcement learning.

I am also curious why the Nearest Neighbor baseline performs so poorly in Table 3. Do the Nearest Neighbor algorithm and NeatNet use the same word embeddings? Currently, it seems like the Nearest Neighbor algorithm places the computer where the desktop was because “desktop” is the most similar word. If so, wouldn’t expanding k, the # of nearest neighbors, help alleviate this problem?

Providing some examples of distinct user preferences in the environments (especially dining and office scenes) would be helpful. Since there are established norms in the non-abstract environments (e.g., keyboard and mouse go in front of the monitor, plate goes in between fork and knife), I’m curious to see how varied these preferences can get. For instance, is the main difference in preferences just left vs right-handedness? Adding some statistics on the variance of user preferences in each of these scenes would help the reader get a better sense of the difficulty of the problem.

Including some visual examples for each of the experiments (especially Table 4) would be great :-)

Please report standard deviation for all means in all the tables.


**Summary Of Recommendation:**

I vote for a weak accept conditioned on the fact that the authors are able to provide stronger baselines in their experiments.

EDIT: Thank you for your reply. I have read it and am raising my score.

---

> ### Author Response · Authors · 2021-08-31
> **Response to Reviewer Ftgd [Part 3 of 3]**
>
> [Part 3 of 3]
>
> **Can Inverse Reinforcement Learning be used for this problem?**
>
> Inverse Reinforcement Learning is an exciting direction for building on this work: it can use our techniques of learning from scenes with Graph Attention Networks and generalising to new objects with word embeddings. An advantage of IRL is that it can balance tidiness with manipulation required, but a disadvantage is that the learning process can be noisier and less efficient than just predicting the tidy state directly, as we do in our approach.
> Our method is the first to learn personalised spatial preferences, but we look forward to future work which creates different approaches and compares them.
>
> **How varied can preferences get?**
>
> We chose left/right-handedness as a running example because it is easy to interpret. Spatial preferences are composed of many other complex factors which are harder to describe, which is why we modelled them with a VAE latent space.
>
> Although most preferences are abstract and difficult to interpret, there are a few patterns we observed looking at some example submissions. For example, users who placed their keyboard near the edge of the table often did the same with their dining cup, suggesting that they were concerned most with reachability. It also tended to be the case that users who thought that compact dining arrangements looked neat (e.g. knife and fork closer together) also arranged other objects compactly (e.g. notepad and pencil). Then there was also significant variation in how users placed their desk lamp, coffee mug, etc. In the abstract scenes, users chose between arranging by colour or by shape, and how close or far apart to space the objects: these preferences persisted across scenes. For objects where preferences do not play a large role, the prior learned by the network from all the training users dominates, and the predicted arrangement still looks tidy.
>
> While this work just introduces the technique of learning spatial preferences, we are keen to see future work which extends this to larger scenes and interprets this further. Our use of Graph Attention Networks means that you can study which object-to-object relations are deemed most important for tidiness by the network by looking at the attention weights. The VAE latent space can even be used to “imagine” entirely new users, or change an existing user’s characteristics to see how their room layout would change.
>
> **Reporting standard deviation in tables**
>
> Thank you, this is important: we have updated the paper accordingly. When user ratings are averaged, the standard deviation across different users is now shown. When the score across multiple scenes is averaged, then we show the mean score across those scenes and the associated standard deviation. When there are only two scenes in an experiment, we present the average only so that methods can be ranked in the table, but the cross-scene standard deviation is omitted as aggregate statistics across two scenes are less meaningful. However, the standard deviations for user scores in each scene are still shown.
>
> **Additionally, note that users are shown all the methods side-by-side before being asked to rank and score them, rather than just giving a rating to one image independently in isolation.** Therefore we also have ranking information which allows us to make conclusions with more confidence than just looking at the means and standard deviations alone. **In a direct comparison, all test users ranked NeatNet above the baselines in Tables 3 & 4**, and all users ranked NeatNet above the Pose-Graph method in Table 1. This means that although user ratings will always have inherent variance due to subjectivity, which is an intrinsic difficulty of the problem being addressed, we can still meaningfully conclude that NeatNet consistently generates tidy and personalised arrangements which are rated highly by users across a variety of rearrangement scenarios.

---

> ### Author Response · Authors · 2021-08-31
> **Response to Reviewer Ftgd [Part 2 of 3]**
>
> [Part 2 of 3]
>
> **Adding more baselines, especially those which are also conditioned on user preferences**
>
> In response to reviewer suggestions, we made 3 major additions to the paper to strengthen the comparison against baselines:
>
> 1. **We developed a new, stronger method called “Weighted-kNN-Regression” for our 9th baseline**, based on the kNN idea suggested in this review. This baseline considers user preferences, because if the user arranges those k objects differently, then this method will produce a different prediction. We arranged individual interviews with the test users, asking them to compare this baseline and other methods in the task of placing new objects: the results are in Table 3. The Weighted-kNN-Regression baseline has a high score on the office scene which comes closer to NeatNet (7.80 vs 9.16), but NeatNet still performs better overall, as discussed on Line 232.
> 2. **We designed and implemented a 10th baseline method called “kNN-Scene-Projection”.** To predict an arrangement for a new scene, kNN-Scene-Projection projects the user's preferences from the example scene onto the new scene, by placing each object in the new scene based on how the user placed the k most similar objects in the example scene. We asked users to rate this method alongside others: the results are in Table 4. Although kNN-Scene-Projection is stronger than all other baselines in this task, NeatNet still performs best, as discussed on Line 239.
> 3. **We added a 4-page description of our Pose-Graph baseline** in the Supplementary Material document. This is a sophisticated model-based method. It learns a probabilistic pose graph representing general tidying preferences using well-established density estimation techniques. The relative constraints in the pose graph are then optimised using a multi-modal pose graph optimisation technique from SLAM literature. This baseline produces neat arrangements, scored highly by users, but NeatNet beats it because it produces personalised arrangements. This comparison demonstrates the value in learning spatial preferences, which is our core contribution.
>
> This means that we now compare against **10 different baselines that we implemented, including 5 which make predictions dependent on preferences** (Weighted-kNN-Regression, Nearest-Neighbour, kNN-Scene-Projection, Positive-Example, Mean-With-Offset) and an ablation baseline (NeatNet-No-Prefs). These comparisons consistently show that NeatNet produces tidy arrangements rated highly by users across many scenarios.
>
> Since NeatNet is the first method to learn personalised spatial preferences, there are no existing methods which have the same capabilities that we could compare against (e.g. generalisation to unseen objects), hence the use of ad-hoc baselines. Future work that develops alternative methods can conduct further experiments on comparing different approaches.
>
> Note that NeatNet is the only method which has all these capabilities: tidying known scenes, placing missing objects, generalising to unseen objects, and transferring to new scenes. Individual baselines can of course be hand-crafted to approach NeatNet’s performance in a specific scenario, but our approach generalises across many rearrangement scenarios, which makes it more suitable for a robot that is useful in the real world. Autonomously predicting a tidy goal state is a crucial capability in robotic rearrangement tasks, making this a useful contribution to the field.

---

> ### Author Response · Authors · 2021-08-31
> **Response to Reviewer Ftgd [Part 1 of 3]**
>
> [Part 1 of 3]
>
> Thank you for your insightful comments and suggestions. Please find our responses below. We have also updated our paper, with additions highlighted in blue.
>
> **Do you have some visual examples of qualitative results?**
>
> Agreed, it would be ideal to include more visual examples in the main paper! This was the case in an early draft, but we moved these figures to the Supplementary Material due to space constraints. If you are interested: feel free to take a look at the qualitative results in the Supplementary Material document on our anonymous project site:
> https://sites.google.com/view/tidying-preferences
>
> We visualise 7 example arrangements generated by various methods, to aid the discussion in the main paper (including for Table 4).  This includes an extra visual comparison we added for the new kNN-Scene-Projection baseline.
>
> **Do the Nearest Neighbour algorithm and NeatNet use the same word embeddings?**
>
> Yes, that is correct - they use the same word embeddings.
>
> **Can you add kNN as a baseline which is more powerful than Nearest-Neighbour?**
>
> kNN is a great idea! Thank you. As requested, we have developed a new, stronger baseline method called “Weighted-kNN-Regression” and asked users to compare it to the other methods. To place a new object, this method finds the k most similar objects in the word embedding space which the user has already arranged. Then it aggregates the positions of those k objects to predict a position for the new object. This aggregation is weighted so that semantically similar objects contribute more to the placement of the new object. This baseline considers user preferences, because it is conditioned on how the user arranges those k objects.
>
> The Weighted-kNN-Regression baseline performs much better than the Nearest-Neighbour baseline on the Office scene (1.70 vs 7.80) by placing the laptop on the table, and has a score which approaches NeatNet (7.80 vs 9.16). One possible reason why NeatNet performs slightly better is that it trains end-to-end with a semantic embedding extractor (Figure 6) which can extract the word vector features that are most relevant to tidying household objects. This is more powerful than just using word embeddings directly: for example, the word “mouse” can mean an animal, but NeatNet can learn to extract the “computer mouse” meaning because it leads to better predictions during training. This means that it can place the laptop in a more convenient way, whereas kNN does not see the mouse as being very relevant to the laptop.
>
> Additionally, NeatNet outperforms Weighted-kNN-Regression on the abstract scene. Consider Figure 2 of the Supplementary Material document (https://sites.google.com/view/tidying-preferences). The task is to place the new item (largest blue box) into the scene. Nearest-Neighbour places it near the large blue box. Weighted-kNN-Regression will place it slightly closer to the centre, actually performing less well than Nearest-Neighbour here, because it breaks the tidy lines created by the user. The NeatNet GNN-VAE correctly learns and extrapolates the pattern made by the user (ascending order of size) to place the new object. Of course, this example is abstract, but it still demonstrates an important spatial reasoning capability: for example, stacking plates or books in order of size would be a common scenario for a household robot.

---

### Official Review · Reviewer_8rCv · 2021-07-23

**Originality:** Very Good
**Technical Quality:** Good
**Clarity Of Presentation:** Excellent
**Impact:** 3

**Recommendation:**

Weak Accept: I recommend accepting the paper, but will not argue for my recommendation if the majority of other reviewers have a different opinion.

**Summary:**

The paper tackles the scene rearrangement task by proposing a novel architecture that learns low-level representations of user preferences in regards to tidying up a space. These representations can subsequently be used to generate object arrangements according to the specific user's spacial preferences and further generalize to include unseen objects through the use of word embeddings. Towards achieving the above, the paper also presents a simulator used for data collection through user interaction. Results receive positive evaluation by users in terms of matching their spatial preferences.

**Issues:**

Ln 15. "Given goal states stipulating where each object should go, many approaches exist for planning a sequence of actions to manipulate objects into the desired arrangement" was followed by only one reference. Could you please cite a few more approaches?

In the related work, an overview of overall existing work on scene rearrangement was expected.

Could the dataset described in https://arxiv.org/abs/2103.16544 be potentially used for further evaluation of the method (instead of or complementary to the developed simulator) as it provides a wider set of scenes? It is very recent work, therefore not expected to have been used in this paper but perhaps it could be useful for future work?

In regards to rearrangement data collection from users, why did you have to build your own simulator? There are many simulators out there, for embodied AI and scene rearrangement.

Ln 19. "Several factors involved in spatial preferences are universal": Please consider rephrasing. One of the two universality examples you give right after, happens to not apply to me. Also, especially since it is subsequently mentioned that "It is clear that spatial preferences are a complex, multi-faceted, and subjective concept" (ln 30). Subjectivity is quite contrary to universality.

Ln 78. "secondly because our method applies prior knowledge from other users": why is this desired behavior? User preferences should be unique vectors in the latent space i.e. other users' preferences should not matter when tailoring a scene to a specific user?

Semantic embeddings section was a bit too brief. Unclear how the embeddings loaded from FastText are tied to the visual scenes and thus, allow for generalization to new objects. Are all objects in a scene labelled by the simulator somehow? Could you please add some more details?


**Reviewer Expertise:**

Good: General knowledge of the area

**Strengths And Weaknesses:**

The paper was very well written. The structure and language significantly helped to follow and comprehend the work done. The combination of VAEs and GNNs was an interesting approach towards learning user preferences and scene representation. Furthermore, generalization to new objects is an important aspect of the problem and the paper did not leave this unaddressed. Combining images with word embeddings for objects in order to address the generalization, was creative.

In terms of weaknesses, one general philosophical question stood out: Important spatial preferences such as left/right handedness are very few and represent a much more constrained space of a user's rationale behind the way they prefer their space organization. For instance, there are people that left/right handedness doesn't necessarily play a part in how they organize their space and/or they are not consistent in a particular location for objects in their space. Secondly, rating organized scenes by humans is not entirely convincing as a good measure of how well the approach works, i.e. due to the inherent subjectivity of user preferences, the same user could be giving different ratings on the same scene at different points in time.


**Summary Of Recommendation:**

Overall, the paper presented an interesting line of work. The problem of user spatial preferences and their use for scene arrangement is inherently difficult due to its subjectivity. The work conducted employed a creative approach that yielded notable results. Along with the paper being well written and easy to follow, it made for a captivating read. In terms of impact, it's not entirely convincing that it can spawn novel directions in the community. However, it can potentially spawn very interesting conversations about the work itself as well as the philosophy of human preferences and thus, would make a solid addition to CoRL 2021.

---

> ### Author Response · Authors · 2021-08-31
> **Response to Reviewer 8rCv [Part 3 of 3]**
>
> [Part 3 of 3]
>
> **What novel directions can this spawn in the community?**
>
> We show that spatial preferences can be learned using VAEs, to determine what the tidy goal state should be for the canonical robotic rearrangement task. Tailoring robot behaviour to account for user preferences opens up many possibilities. Future work can add this personalisation layer to other tidying methods, apply personalisation in collaborative settings, and balance tidiness with time efficiency. The VAE latent space can even be used to “imagine” entirely new users, or change an existing user’s characteristics to see how their room layout would change.
>
> Although our core contribution is learning spatial preferences, we also contribute a novel way of modelling scenes of objects with Graph Attention Networks, and generalising to new objects using word embeddings, which will also be of interest to the robot learning community.
>
> **Could this dataset from a recent paper be used as part of future work with this method?**
>
> Yes, this is very helpful - thank you! It seems like a very wide range of realistic scenes and objects. This will be useful to build up a strong prior about how users in general prefer to place certain objects, and then we can apply user-specific personalisation on top of this. Each scene would be treated as if it were arranged by a different training user. It might be necessary to have a human verify that only tidy generated arrangements are used, although they look very reasonable from the examples in that paper. We would still need a dataset of human users who personally arrange several scenes: this allows the network to learn how preferences transfer across scenes arranged by the same user. After combining the datasets, the training algorithm will proceed as before.
>
> **Why did you build your own simulator?**
>
> There are indeed many simulators out there already, with powerful features designed for AI/Robotics experts. Our simulator has a different purpose: crowd-sourced, large-scale data collection. The simulator is: (1) Deployed as a web app, so that people can see the post on social media and join the study immediately; (2) Very easy to use for ordinary people who have no prior experience of 3D rearrangement simulators; (3) Connected to a cloud database back-end so that data is effortlessly submitted for experiments; (4) Tailored to rearrangement tasks, e.g. with an inventory system, and multiple pages of scenarios to be completed. Our simulator code is available with the paper and, if published, will be useful for robotics researchers who want to gather user-created rearrangement data on a larger scale for creating deep learning models that can learn spatial preferences. This complements existing simulators, because the purpose is slightly different.
>
> **Why is using prior knowledge from other users desired behaviour?**
>
> To confirm: every user gets unique, personalised predictions. Prior knowledge from observing training users is stored in the learned weights of the neural network. This is essential for generalisation. Imagine that a test user acquires a new object. The network does not know how they want to place it. But if the network has seen a training user with “similar” preferences place that object before, it can make a reasonable prediction for how the new user wants that object to be placed. This is analogous to how Amazon or YouTube leverage large-scale data about other users while still giving you personalised predictions. Alternatively, think of a VAE used for correcting damage to a picture of a face: although it has never seen that exact face before, it has a general idea of what faces can look like, so it can fill in the gaps using that prior knowledge about “similar” faces. We have rephrased that line to clarify this idea: “...and secondly because our method combines knowledge about the current user’s preferences with prior knowledge about “similar” training users, correcting errors in individual examples and allowing for stronger generalisation” (Line 81).

---

> ### Author Response · Authors · 2021-08-31
> **Response to Reviewer 8rCv [Part 2 of 3]**
>
> [Part 2 of 3]
>
> **Why were user ratings chosen as the main evaluation metric for personalised arrangements generated by each method?**
>
> We agree that subjectivity is a downside of any user ratings metric, since it makes it harder to obtain results. As the reviewer mentions, subjectivity is inherent to the challenging problem being addressed.
>
> A robot’s ultimate goal is to make users happy. If a method makes users happy, then it is a successful method: therefore, user ratings of arrangements is the most direct success metric available. Timewise fluctuations of user ratings is also a fair point. Future work might take multiple ratings from the same user over a period of time to account for this.
>
> It is possible to use a proxy metric such as the distance error between a predicted position and some ground truth position supplied by a user. In fact, we also provide the results of an experiment which does use this objective metric in Table 2, because it is more precise than user ratings.
>
> However, we chose user ratings as the predominant metric because distance error could give misleading results. For example, there might be multiple positions of a mug on a table deemed equally acceptable to the user, whereas the distance error metric assumes that there is one objectively correct target position (or region). Alternatively: suppose that the user places their coffee mug on the dining table, but is also fine with it being on the kitchen counter. Method A places the mug on the kitchen counter. Method B places the mug on the floor between the two surfaces. Now Method B would win on distance error, even though Method A is strongly preferred by the user! So using a distance metric might in some cases *over-represent* the success of a method, whereas user ratings are a more direct and legitimate measure of success.
>
> Another evaluation method which we used in the early stages of this research involved designing artificial humans and programming their ground-truth preferences. However, modelling something as complex as spatial preferences would force many assumptions, and so there is more research value in working with data from real human users, even if it is more difficult.
>
> **How important are preferences, beyond left/right handedness?**
>
> We chose left/right handedness as a running example because it is easy to interpret. Spatial preferences are composed of many other complex factors which are harder to describe in words, such as aesthetic beauty, convenience, safety, etc. This complexity is exactly why a VAE latent space is suitable for modelling spatial preferences. Convenience in itself is highly subjective: two random people would likely organise the same fridge differently because they have different usage patterns for ingredients. These subjective factors also greatly influence how people tidy their shelves, clothes, desks, etc: there is huge diversity in spatial preferences, and they often play an important (but difficult to describe) role in many arrangements. We also show experimentally that taking preferences into account improves tidying performance. Since our method is the first to learn personalised spatial preferences, it will be a useful starting point for future work to investigate preferences more deeply in more complex scenes, where there will be even more variance due to subjectivity.
>
> **People are not consistent in a particular location for objects in their space**
>
> It is true that there are probably many possible arrangements which are considered acceptable for a user. The robot does not have to predict or model every acceptable arrangement, but just needs to predict one acceptable tidy arrangement to complete its task. Our method also makes it easy to handle preferences that evolve over time: the user can just delete old example arrangements and add new examples, which will alter the inferred user preference vector.

---

> ### Author Response · Authors · 2021-08-31
> **Response to Reviewer 8rCv [Part 1 of 3]**
>
> [Part 1 of 3]
>
> Thank you for your insightful comments and suggestions. Please find our responses below. We have also updated our paper, with additions highlighted in blue.
>
> **Preferences are subjective and not universal**
>
> The reviewer is correct. We have now rephrased this. Spatial preferences are indeed subjective and no factors can be described as universal.
>
> The idea here is that although every user represents a unique combination of preferences, there are a multitude of factors which are considered by many users, e.g. symmetry, convenience, safety/stability, etc. Therefore it is indeed possible to discover some latent patterns in user preferences, even if each individual is unique (similar idea to neural recommenders used by Amazon/Netflix/YouTube etc - see [13,14] in paper). We have updated the paper to reflect this: “Several factors involved in spatial preferences are shared across many users: symmetry and usefulness are preferable to chaos.” (Line 19).
>
> **Are all objects in a scene labelled by the simulator?**
>
> Yes, object names are provided by the simulator. If deployed physically, NeatNet would be one part of a larger robotic pipeline: existing approaches can be used for localisation, object detection and pose estimation [5-7], which are all active and independent areas of research. These methods would determine the names and poses of objects in a scene and pass this data on to NeatNet. The crucial role filled by our method is predicting a tidy arrangement for this set of detected objects, which is also important for a tidying robot to be useful. We have now added an answer to this question to the semantic embeddings section: “Names of objects are provided to NeatNet either by the simulator or by an object detection system if deployed physically” (Line 155).
>
> **Could you please cite a few more approaches to planning?**
>
> Yes, we would be happy to. The paper we cited is a survey paper on the task of rearrangement, including an overview of planning approaches. We did not focus on any specific planning method since NeatNet can work with any planning method. However, we have now cited two further survey papers to provide a broader perspective on the field (Line 16).
>
> **Overview of existing work on scene rearrangement**
>
> Thank you for your suggestion: we agree that adding this overview has improved the presentation of the Related Work section. Our previous draft dived too quickly into discussing specific approaches for scene rearrangement and comparing them against ours. We have now updated the section to include an introductory paragraph containing a broader overview of 3 different types of approaches, before discussing specific methods in more detail. Furthermore, our Introduction section includes a discussion of where our method fits into a robotic pipeline alongside localisation, object identification, pose estimation and planning (Lines 30-40).

---

### Official Review · Reviewer_vXaq · 2021-07-26

**Originality:** Excellent
**Technical Quality:** Very Good
**Clarity Of Presentation:** Very Good
**Impact:** 3

**Recommendation:**

Weak Accept: I recommend accepting the paper, but will not argue for my recommendation if the majority of other reviewers have a different opinion.

**Summary:**

The main contributions of this paper could be concluded as following:
(1). Proposed a GNN-based VAE model named NeatNet to deal with rearrangement tasks, which can extract more latent information like the spatial information of input scene images
(2). Innovatively bring out the new form of rearrangement tasks: rearrange the given scene into personalized style with preference extracted from provided train samples
(3).Gathered preference samples from users and validate the effectiveness of proposed model with experiments


**Issues:**

1. More realistic and complex samples should be included
2. A more complicate and effective baseline should be used in comparison to make the results more convincing


**Reviewer Expertise:**

Good: General knowledge of the area

**Strengths And Weaknesses:**

Strengths:
1.	Innovatively proposed the method that use Graph Neural Network layers in VAE, a traditional generative model which may allow the model directly learn from arrangement provided by users in format of graphs
2.	Designed a well-considered attention mechanism and update rules to make the model extract the most important preference in samples.
3.	Comprehensively explore the performance of propose model in different situations with conducting experiments, such as generate arrangements for known scenarios, fit a new kind of object in current scenario and generalize a learned preference to a new scenario.
Weakness:
1.	More realistic and complex scenarios need to be used to validate the effectiveness and generalizability of proposed model. For reality, images of some real scenarios with same arrangements should be concluded. For complexity, scenarios with larger views such as a whole scene of dining room instead of just the table.
2.	Baselines should be more convincing instead of only naive methods implemented in experiment.


**Summary Of Recommendation:**

I recommend this paper for its contributions on forming a innoative type of robot re-arrangement tasks and proposing the GNN-based  VAE model to deal with such kind of rearrangement tasks. The GNN-based model allow the generative model  directly learn from arrangement provided by users in format of images and extract the preference information from latent space. And the experiments validated its effectiveness with comparison to several naïve baselines.

---

> ### Author Response · Authors · 2021-08-31
> **Response to Reviewer vXaq [Part 2 of 2]**
>
> [Part 2 of 2]
>
> **Have you compared this approach against effective baselines?**
>
> In response to reviewer suggestions, we made 3 major additions to the paper to strengthen the comparison against baselines:
>
> 1. **We developed a new, stronger method called “Weighted-kNN-Regression” for our 9th baseline.** To place a new object, this method finds the k most similar objects in the word embedding space which the user has already arranged. Then it aggregates the positions of those k objects to predict a position for the new object. This aggregation is weighted so that semantically similar objects contribute more to the placement of the new object. This baseline considers user preferences, because it is conditioned on how the user arranges those k objects. We arranged individual interviews with the test users, asking them to compare this baseline and other methods in the task of placing new objects: the results are in Table 3. The Weighted-kNN-Regression baseline has a high score on the office scene which comes closer to NeatNet (7.80 vs 9.16), but NeatNet still performs better overall, as discussed on Line 232.
> 2. **We designed and implemented a 10th baseline method called “kNN-Scene-Projection”.** To predict an arrangement for a new scene, kNN-Scene-Projection projects the user's preferences from the example scene onto the new scene, by placing each object in the new scene based on how the user placed the k most similar objects in the example scene. We asked users to rate this method alongside others: the results are in Table 4. Although kNN-Scene-Projection is stronger than all other baselines in this task, NeatNet still performs best, as discussed on Line 239.
> 3. **We added a 4-page description of our Pose-Graph baseline** in the Supplementary Material document. This is a sophisticated model-based method. It learns a probabilistic pose graph representing general tidying preferences using well-established density estimation techniques. The relative constraints in the pose graph are then optimised using a multi-modal pose graph optimisation technique from SLAM literature. This baseline produces neat arrangements, scored highly by users, but NeatNet beats it because it produces personalised arrangements. This comparison demonstrates the value in learning spatial preferences, which is our core contribution.
>
> This means that we now compare against **10 different baselines that we implemented, including 5 which make predictions dependent on preferences** (Weighted-kNN-Regression, Nearest-Neighbour, kNN-Scene-Projection, Positive-Example, Mean-With-Offset) and an ablation baseline (NeatNet-No-Prefs). These comparisons consistently show that NeatNet produces tidy arrangements rated highly by users across many scenarios.
>
> Since NeatNet is the first method to learn personalised spatial preferences, there are no existing methods which have the same capabilities that we could compare against (e.g. generalisation to unseen objects), hence the use of ad-hoc baselines. Future work that develops alternative methods can conduct further experiments on comparing different approaches.
>
> Note that NeatNet is the only method which has all these capabilities: tidying known scenes, placing missing objects, generalising to unseen objects, and transferring to new scenes. Individual baselines can of course be hand-crafted to approach NeatNet’s performance in a specific scenario, but our approach generalises across many rearrangement scenarios, which makes it more suitable for a robot that is useful in the real world. Autonomously predicting a tidy goal state is a crucial capability in robotic rearrangement tasks, making this a useful contribution to the field.

---

> ### Author Response · Authors · 2021-08-31
> **Response to Reviewer vXaq [Part 1 of 2]**
>
> [Part 1 of 2]
>
> Thank you for your insightful comments and suggestions. Please find our responses below. We have also updated our paper, with additions highlighted in blue.
>
> **What would you need to do to apply this in the physical world?**
>
> If deployed physically, NeatNet would be one part of a larger robotic pipeline: existing approaches can be used for localisation, object identification, pose estimation, planning, grasping, etc. (see [1-3, 5-7] in paper), which are all active and independent areas of research. NeatNet fills the crucial role of high-level reasoning about what the target state of the rearrangement should be. We would of course be excited to demonstrate our method in the real world as future work. Whereas existing methods rely on near-perfect positive examples [12], our method is able to learn from imperfect examples since the VAE corrects errors using learned prior knowledge (Table 1), which makes it more likely to transfer well to the physical world.
>
> **Can this approach scale up to larger scenes?**
>
> The complexity of scenes used in these experiments is limited by the requirement to get crowd-sourced data from as many users as possible, since the focus of this paper is on subjective preferences. We found that users from social media were unwilling to spend much more than 5 minutes making arrangements, so we designed scenes which maximised experimental value while keeping user effort low. However, we still obtained data on 4 different scenes with 30 distinct objects (abstract and real) used in experiments. The complexity in this challenge comes from learning personalised preferences from 75 different users, a significant advance on prior work [11] which has 5 users and does not demonstrate personalised predictions. Since ours is the first method to learn personalised spatial preferences, it is a starting point for future work which could include investigation into more complex scenes. Since Graph Neural Networks are powerful models which are proven to work with thousands of nodes [19], we have good reason to believe that this approach is likely to scale well to larger scenes.

---

### Meta-Review · Area_Chair_2NNA · 2021-08-13

**Recommendation:** Accept (Poster)
**Confidence:** 5

**Metareview:**

In this paper, the authors propose the novel NeatNet considering user preference for robotic rearrangement task. The variational autoencoder is used to learn a low-dimensional preference vector and the scenes are modeled with GNN. All reviewers recognized that the paper is well written and organized. It will make the paper more convincing if some stronger baselines, for example, considering user preference, are provided. Also, more qualitative results are expected to make the experimental results more illustrative.

In the rebuttal session, the authors have responded to the reviewers' concerns and more details are provided. And the reviewers have reached a consensus of accepting this paper.

---

> ### Author Response · Authors · 2021-08-31
> **Response to Meta Review [Part 2 of 2]**
>
> [Part 2 of 2]
>
> Reviewer vXaq requested “more complicated and effective baselines”. This was also given as a condition by Reviewer Ftgd for their recommendation. **We made 3 major additions to the paper to meet this condition:**
> 1. **We developed a new, stronger method called “Weighted-kNN-Regression” for our 9th baseline**, based on a suggestion made by Reviewer Ftgd. To place a new object, this method finds the k most similar objects in the word embedding space which the user has already arranged. Then it aggregates the positions of those k objects to predict a position for the new object. This aggregation is weighted so that semantically similar objects contribute more to the placement of the new object. This baseline considers user preferences, because it is conditioned on how the user arranges those k objects. We arranged individual interviews with the test users, asking them to compare this baseline and other methods in the task of placing new objects: the results are in Table 3. The Weighted-kNN-Regression baseline has a high score on the office scene which comes closer to NeatNet (7.80 vs 9.16), but NeatNet still performs better overall, as discussed on Line 232.
> 2. **We designed and implemented a 10th baseline method called “kNN-Scene-Projection”.** To predict an arrangement for a new scene, kNN-Scene-Projection projects the user's preferences from the example scene onto the new scene, by placing each object in the new scene based on how the user placed the k most similar objects in the example scene. We asked users to rate this method alongside others: the results are in Table 4. Although kNN-Scene-Projection is stronger than all other baselines in this task, NeatNet still performs best, as discussed on Line 239.
> 3. **We added a 4-page description of our Pose-Graph baseline** in the Supplementary Material document. This is a sophisticated model-based method. It learns a probabilistic pose graph representing general tidying preferences using well-established density estimation techniques. The relative constraints in the pose graph are then optimised using a multi-modal pose graph optimisation technique from SLAM literature. This baseline produces neat arrangements, scored highly by users, but NeatNet beats it because it produces personalised arrangements. This comparison demonstrates the value in learning spatial preferences, which is our core contribution.
>
> This means that we now compare against **10 different baselines that we implemented, including 5 which make predictions dependent on preferences** (Weighted-kNN-Regression, Nearest-Neighbour, kNN-Scene-Projection, Positive-Example, Mean-With-Offset) and an ablation baseline (NeatNet-No-Prefs). These comparisons consistently show that NeatNet produces tidy arrangements rated highly by users across many scenarios.
>
> Since NeatNet is the first method to learn personalised spatial preferences, there are no existing methods which have the same capabilities that we could compare against (e.g. generalisation to unseen objects), hence the use of ad-hoc baselines. Future work that develops alternative methods can conduct further experiments on comparing different approaches.
>
> Note that NeatNet is the only method which has all these capabilities: tidying known scenes, placing missing objects, generalising to unseen objects, and transferring to new scenes. Individual baselines can of course be hand-crafted to approach NeatNet’s performance in a specific scenario, but our approach generalises across many rearrangement scenarios, which makes it more suitable for a robot that is useful in the real world. Autonomously predicting a tidy goal state is a crucial capability in robotic rearrangement tasks, making this a useful contribution to the field.
>
> We have also made several additional changes in response to reviewer comments: the details are in the responses to each reviewer.

---

> ### Author Response · Authors · 2021-08-31
> **Response to Meta Review [Part 1 of 2]**
>
> [Part 1 of 2]
>
> We would like to thank the Area Chair for their feedback, and indeed all the reviewers for their insights and suggestions. We have updated our paper accordingly, with additions highlighted in blue.
>
> Our core contribution is NeatNet, an architecture for learning spatial preferences. Scenes are modelled using Graph Neural Networks. A Variational Autoencoder is used to extract a latent user preference vector from example scenes. This vector is used to generate personalised tidy arrangements. We are delighted that the reviewers found this approach “novel” (Reviewer Ftgd)  and “innovative” with “excellent” originality (Reviewer vXaq), addressing an “inherently difficult” problem (Reviewer 8rCv). Word embeddings are used to generalise to new objects. Reviewer 8rCv described this as a “creative” way to address an “important aspect” of the problem. We developed a simulator to gather user-made rearrangement data on a large scale. NeatNet was evaluated in its ability to tidy known scenes, place missing objects, generalise to new objects, and predict arrangements for new scenes. It consistently generated neat and personalised arrangements, rated highly by users. The experiments included 10 baselines that we developed (including those requested by reviewers), using 30 objects across 4 scenes. The reviewers found the evaluation experiments to be “comprehensive” (Reviewers vXaq and Ftgd), yielding “notable results” (Reviewer 8rCv). Furthermore, we are glad that the reviewers found the paper “clearly written and organised” (Reviewers Ftgd and vXaq), making it a “captivating read” (Reviewer 8rCv). This is the first method to learn personalised spatial preferences, which is crucial for household robots to be useful.
>
> Reviewer Ftgd requested to see **visual examples of qualitative results**. We agree that it would be ideal to include these in the main paper. This was the case in an early draft, but we moved these figures to the Supplementary Material due to space constraints. For visual examples, feel free to take a look at the figures in the Supplementary Material document on our anonymous project site:
> https://sites.google.com/view/tidying-preferences
>
> We visualise 7 example arrangements generated by various methods across different experiments, to aid the qualitative discussion in the main paper. This includes an extra visual comparison we added for the new kNN-Scene-Projection baseline, in response to reviewer comments (Supplementary Material Figure 4).

---

### Decision · Program_Chairs · 2021-09-13

**Decision:**

Accept (Poster)

**Comment:**

In this paper, the authors propose the novel NeatNet considering user preference for robotic rearrangement task. The variational autoencoder is used to learn a low-dimensional preference vector and the scenes are modeled with GNN. All reviewers recognized that the paper is well written and organized. It will make the paper more convincing if some stronger baselines, for example, considering user preference, are provided. Also, more qualitative results are expected to make the experimental results more illustrative.

In the rebuttal session, the authors have responded to the reviewers' concerns and more details are provided. And the reviewers have reached a consensus of accepting this paper.